# Efficient sampling of large-scale transition pathways and intermediate conformations in sub-mesoscopic protein complexes

Domenico Scaramozzino [ID] , Byung Ho Lee [ID] & Laura Orellana [ID] [✉]

Protein conformational changes are the cornerstone of biological function. While conformers captured experimentally represent metastable states, the pathways connecting them have been elusive for experiments and simulations alike. Nowadays, cryogenic Electron Microscopy is providing rich structural data on proteins trapped in different states for increasingly large systems, but these are out of scope for most computational methods which exhibit an $N^2$ dependence on size. Based on our previous eBDIMS algorithm, here we present eBDIMS2, an optimized version with quasi-linear size dependence, able to simulate on a desktop computer particularly complex transitions for megadalton protein assemblies, like the rotary motion of ATP synthases. Not only eBDIMS2 pathways spontaneously visit experimental intermediates but also overlap with enhanced and microsecond Molecular Dynamics simulations requiring extensive supercomputing resources. By integrating Elastic Networks with Brownian Dynamics, eBDIMS2 allows an unprecedented exploration of conformational changes of sub-mesoscopic systems previously inaccessible.

Protein dynamics is the fundamental link between structure and function[1–3], refined and conserved through evolution[4]. In response to signals such as electrochemical gradients or ligand binding, proteins transition between different states or conformations—such as open/closed, active/inactive, etc. —enabling biological regulation. Understanding these processes requires bridging static experimental snapshots to uncover transition pathways for conformational changes[5], often involving transient intermediates critical for function[6]. Despite advances in hardware and software[7], sampling such transitions with Molecular Dynamics (MD)[8]—the gold standard for biomolecular simulations—remains challenging for the time and length scales involved[3]. To increase the exploration of the conformational space, numerous methods have been proposed[9]. While some rely on tricks to enhance the atomistic sampling[10,11], others employ coarse-grained (CG) techniques to simplify the physical modeling[12]. However, both approaches require complicated setups and often significant computing resources. As a faster alternative, minimalist CG-methods like the Elastic Network Models (ENMs)[13] can predict conformation

changes with remarkable precision[14,15], leading to numerous variants[16–22] to model protein dynamics[4,23], flexibility[24,25], or guide transition pathways. However, their lack of rigorous CG-parametrization limits transferability and general applicability.

Building on our carefully MD-parametrized edENM force-field[19], we previously developed eBDIMS[26] (elastic network-driven Brownian Dynamics IMportance Sampling), an ENM-based simulation to track transition paths between protein end-states. To assess the biological relevance of the predicted transitions beyond stereochemical accuracy, we followed the approach proposed by Levitt[27], validating the pathways against a benchmark of proteins with experimentally trapped transition states and testing whether such intermediates were spontaneously visited without prior information. To further evaluate the mechanistic coherence of the transitions, we applied principal component analysis (PCA)[28] to the intermediate-containing structural ensembles. This allowed direct visualization of whether the sampled motions explored the experimental conformational space in a sequential and structurally coherent manner. When tested against

Protein Dynamics and Mutation Lab, Department of Oncology-Pathology, Karolinska Institutet, Solna, Sweden. [✉]e-mail: laura.orellana@ki.se

other path-sampling methods, eBDIMS was the only CG method capable of predicting smooth pathways in PC-space with the accuracy of MD or atomistic path-sampling methods, like Climber[27], but at a fraction of the computational cost[26]. The high quality of the predicted intermediates has enabled seeding of full-atom MD simulations to explore the Free Energy Landscape (FEL) of very subtle allosteric transitions, such as ion channel gating[29,30]. Providing mechanistic insights across diverse systems, from transcription factors[31] to enzymes or fusion proteins[32,33], eBDIMS has proven to be a general and versatile method for studying protein transitions. Another key application has been in cryogenic Electron Microscopy (cryo-EM) studies[34,35], where eBDIMS results enabled to explain intermediates between trapped conformations[36].

However, in the past years, the cryo-EM resolution revolution[37] has been dramatically expanding the Protein Data Bank (PDB)[38] with large multi-state protein complexes[39–41], creating challenges for computational methods that often scale quadratically with system size. This has made large-scale transitions in protein assemblies > 300 kDa nearly intractable even for CG methods like eBDIMS, requiring resources that few labs can afford. For instance, observing the opening of the *SARS-CoV-2* spike glycoprotein required enhanced Weighted Ensemble (WE) simulations running on multiple GPUs[42]. Similarly, capturing the conformational variability of the GroEL chaperone demanded hundreds of microseconds on the special-purpose supercomputer Anton[43]. In large multimers like GroEL, most CG path-sampling requires several days of computation to recover a full transition pathway, and the only methods currently capable of handling such systems, MinActionPath2[44] and NOLB[45], are found to produce significant structural distortions or not fully converge to targets (see below).

In this work, we present eBDIMS2 (Fig. 1), an extension of our previous algorithm which leverages cutoff and parallelization schemes to achieve a quasi-linear scaling with the system size. This enables efficient CG-simulations of much larger proteins (up to ~2 MDa) and difficult transitions on a standard computer, with virtually no constraints on system size, architecture, or motion complexity. Here, to evaluate the mechanistic and biological relevance of eBDIMS2, we assembled an updated dataset of large protein ensembles and simulated transitions between all relevant end states. Our results show that eBDIMS2 preserves the ability of the previous version to generate smooth, physically coherent pathways consistent with experimental intermediates and MD, while extending this capability to systems of unprecedented size and complexity. In particular, eBDIMS2 efficiently captures large-scale transitions of exceptional difficulty, such as the activation–deactivation cycle of neurofibromin (Nf1) and the full rotary motion of ATP synthases. As cryo-EM data for such sub-mesoscopic protein complexes continue to grow, eBDIMS2 will provide a powerful and accessible tool for studying the dynamics of these critical yet underexplored systems, beyond the reach of most computational methods.

## Results

### eBDIMS2 large-scale protein motion benchmark and ensemble validation by PCA

To validate the eBDIMS2 path-sampling algorithm, we expanded our previous benchmark composed of proteins that have well-characterized intermediates between end-states (e.g., RBP, RNaseIII, and SERCA)[26] by performing an exhaustive search for larger and conformationally diverse systems in the PDB. This resulted in a total of 47 large proteins of different stoichiometry and motions (Fig. 2a), for which conformationally rich ensembles were retrieved for robust ensemble-PCA (see below). All complexes are larger than ~300 kDa, have at least 3 experimental models available, and Root Mean Square Deviations (RMSDs) between two end states of at least ~4 Å. This resulted in a collection of 872 PDB structures, mostly from cryo-EM,

with ensembles containing from a minimum of 3 to nearly 90 structures, and an average ensemble RMSD over ~7 Å (Supplementary Table 4). The benchmark includes protein assemblies ranging from the ~300 kDa *SARS-CoV* spike glycoprotein to the ~2.3 MDa *S. Cerevisiae* Fatty-Acid Synthase (FAS, Fig. 2a).

Most of the conformational changes in these protein ensembles are large-scale and collective (Fig. 2b), with transition RMSDs ranging from a minimum of ~4 Å in the gigantic ryanodine receptor 2 (RyR2, ~16 k residues), to an astonishing ~30 Å for isoform 1 of neurofibromin (Nf1, ~4.3k residues). Upon PCA (see Methods), the first two PCs of such conformationally-rich ensembles capture >70% of the structural variance in all ensembles, and >90% in ~70% of the cases (Supplementary Table 6), thus providing powerful Collective Variables (CVs) for system analysis, as shown previously by us[26,46] and others[47,48]. PC1 and PC2 typically describe global motions, such as hinge-bending, twisting, and breathing modes (Supplementary Figs. 8–12), and as user-independent intrinsic CVs, they facilitate the identification of main conformers (e.g., open/closed, inward/outward, etc.) and their interconnecting pathways. Using PC projections as reference, we identified 124 relevant end states and simulated 191 transition pathways with eBDIMS2 (Supplementary Table 7), identifying experimental intermediates in >30% of PC spaces.

To further enrich our protein motion dataset, we incorporated 15 additional large protein complexes, each exhibiting two distinct experimentally determined conformations from the PDB (Supplementary Table 5). This expansion increases the diversity of the conformational transitions and structural architectures represented in our protein benchmark. It also underscores the robustness of our updated algorithm in handling even more challenging cases, such as the ~57 Å transition between the two pH-dependent conformations of lipoprotein receptor-related protein 2 (LRP2, ~840 kDa), as well as experimental end states with differing residue and chain compositions, e.g., as observed in the two conformers of the integrator-PP2A complex (~1 MDa).

### eBDIMS2 successfully simulates conformational transitions in large proteins with different architectures and achieves a quasi-linear size-time dependence

For a few full-length proteins, we compared eBDIMS2 to other existing path-sampling algorithms (see Methods). The obtained paths were evaluated in terms of computational speed, target convergence, and smoothness/stability of the sampling in PC projections. While for small- to medium-size proteins (<1k residues), eBDIMS2 shows similar performances compared to other existing algorithms (Supplementary Fig. 16), it stands out when dealing with larger systems (>300 kDa). For example, the ~15 Å transition of GroEL 7-mer (~400 kDa) can be simulated by eBDIMS2 in ~1 h, reaching an astonishing convergence of ~0.6 Å to the target state (Fig. 2d), while the majority of the investigated methods is not able to simulate the full transition in less than 12 h. To achieve the same convergence, our previous algorithm[46] takes ~7 h, highlighting a dramatic ~6-fold performance increase (Fig. 2d). Notably, only NOLB and the recently published MinActionPath2 can handle systems of this size, generating transition frames within minutes. However, NOLB has poor convergence to the target state (RMSD ~10 Å, Fig. 2d), while MinActionPath2 consistently introduces important backbone distortions at the transition points (Supplementary Fig. 17). PC projections reveal that other path-sampling methods often tend to sample the conformational space with abrupt directional changes (seen as zig-zag projections in Fig. 2d). In contrast, eBDIMS and eBDIMS2 consistently trace smoother pathways that follow the natural PC-motions, and moreover, overlap with those obtained with the atomistic method Climber for small- and medium-sized proteins[26]. Apart from speed and sampling stability, eBDIMS2 has also special versatility as it can deal with structures containing missing residues (a necessary condition when dealing with large cryo-EM models) and it

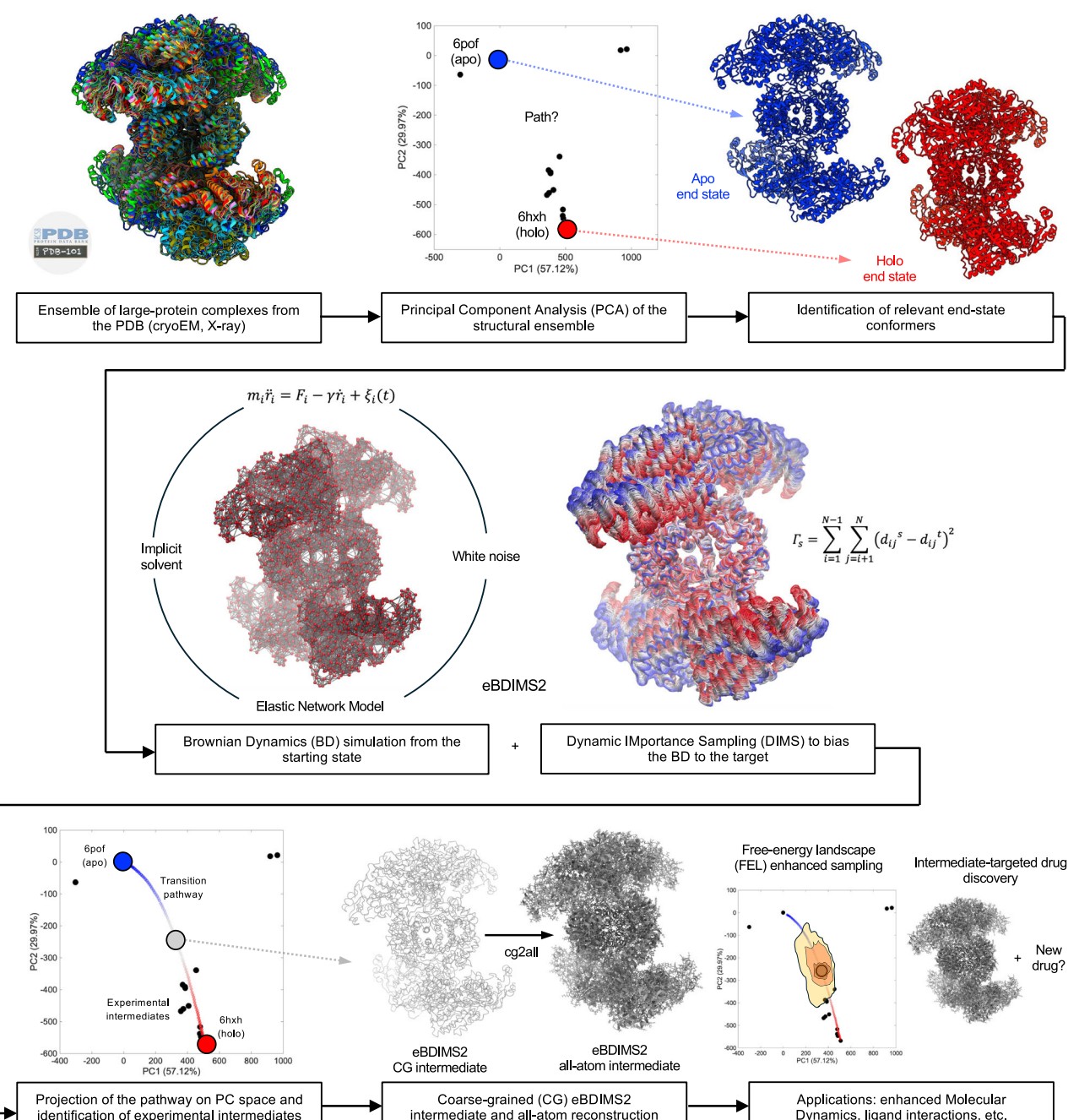

**Fig. 1 | Overview of the eBDIMS2 path-sampling method for the generation of conformational intermediates of large protein systems, cross-validation with Principal Component Analysis (PCA) of experimental ensembles, and applications upon atomistic reconstruction.** The first step is to generate an ensemble from all available structures in the Protein Data Bank (PDB) obtained from X-ray crystallography or cryogenic Electron Microscopy (cryoEM) experiments. PCA is then performed to cluster experimental conformations and assign biological states to each conformational cluster, e.g., apo/holo, inactive/active. After identification of relevant end-state conformers, eBDIMS2 uses a combination of coarse-grained (CG) Elastic Network Modeling (ENM) and Brownian Dynamics (BD) to sample the conformational transition between the two states. The transition pathway is projected back onto the experimental PC space and used to explain experimental intermediates and conformational cycles. The CG intermediates can be atomistically reconstructed and used for downstream applications, e.g., enhanced sampling with Molecular Dynamics (MD), drug design targeting the intermediate conformation, etc.

can also generate convergent paths between conformers with different residue and chain compositions, thus facilitating, e.g., reconstruction of alternate states from full-length ones (Supplementary Fig. 7).

Overall, eBDIMS2 computing times for large systems in our ensemble dataset have a median value of ~2.5 h (Fig. 2b) and range from a minimum of ~19 min for DNA-dependent protein kinase catalytic subunit (DNA-PKcs, ~3k residues, ~6 Å RMSD) to a maximum of ~49 h for the gigantic FAS 12-mer (~21k residues, ~5 Å; transition details in Supplementary Table 7). The transitions are routinely able to reach the target state with high convergence, below ~1 Å in most cases (Fig. 2b, orange bars). As expected, we observe a clear correlation between computing times and system sizes, with a Pearson Correlation Coefficient (PCC) of ~0.9, showing a quasi-linear size-time dependency (Fig. 2c, left panel). Even if we exclude the three largest systems in our ensemble dataset (the two ryanodine receptors and FAS 12-mer, >1.5

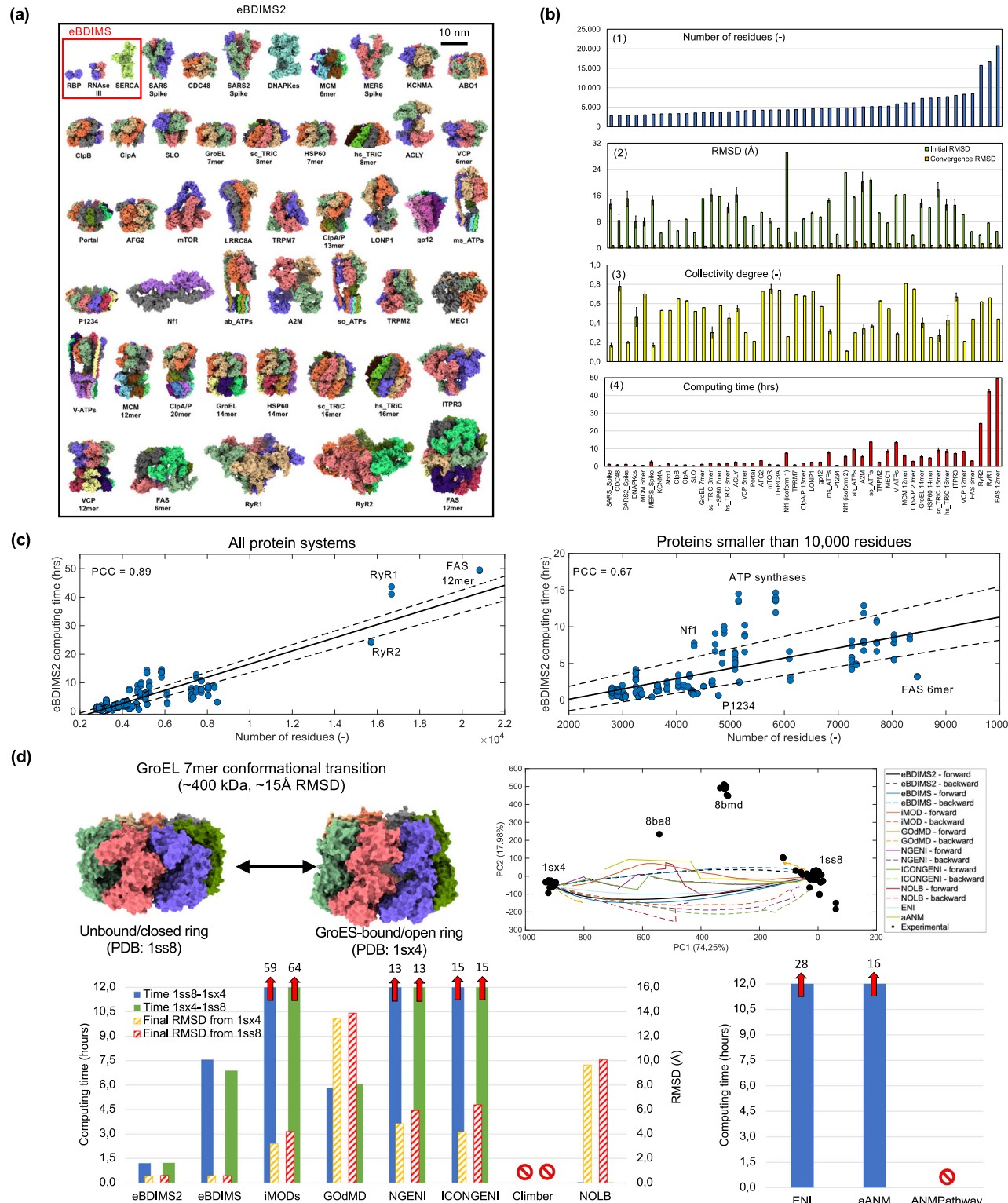

MDa), the linear size-time correlation is still present, with a correlation of ~0.7 (Fig. 2c, right). This confirms the enhanced performance of eBDIMS2 compared with our previous algorithm version and other path-sampling methods that often follow an N² dependence with size.

We also expected computing times not only to depend on the protein size, but also on the extent and complexity of the transition, as can be seen from outliers in the size-time relationship (Fig. 2c, right). Considering two similarly sized systems, we expect that the larger the extent of a transition, i.e., the larger the RMSD between the end states, the larger the computing time. This is confirmed by a positive PCC

(~0.5) between the size-normalized computing times vs. RMSD values (see Supplementary Information). On the other hand, size-normalized computing times also display a slightly negative correlation versus collectivity degrees (~−0.2). This suggests that eBDIMS2 is faster for global-collective conformational changes, likely arising from the tendency of ENMs to model large domain motions better than localized ones[3,14,23]. As an example of two similarly sized systems but with different conformational changes, polyprotein P1234 and isoform 2 of Nf1 (both ~500 kDa) undergo completely distinct motions, thus leading to different outcomes in terms of computing effort (Fig. 2c, right). P1234

**Fig. 2 | eBDIMS2 protein motion benchmark and computing times. a** Large proteins in our ensemble dataset (from ~300 kDa to ~2.3 MDa; Supplementary Table 4). The three medium-sized proteins from our previous eBDIMS benchmark[26] are also shown here for size comparison. **b** Transition pathways in the ensemble dataset simulated with eBDIMS2: (1) system size (number of common residues in the ensemble); (2) initial Root Mean Squared Deviation (RMSD) between end states (green bars) and convergence RMSD (orange), averaged across all analyzed transitions per protein; (3) collectivity degree of the conformational changes; (4) eBDIMS2 computing time to reach convergence. Individual data points for each system, as well as the total number of transitions used to compute average values and standard deviations presented in this plot, can be found in the Supplementary Information and in the Source Data file. **c** Correlation between eBDIMS2 computing time and system size (number of residues), for all 47 large protein systems (left panel, PCC ~ 0.9) and for proteins smaller than ~1 MDa (right panel, PCC ~ 0.7). The black line corresponds to the trend line of the dataset, while the dashed lines are related to 95% confidence intervals. **d** Transition pathways in GroEL 7-mer (~400 kDa) and comparison of computing times between eBDIMS2 and other non-linear (left panel) and linear (right panel) path-sampling methods. For non-linear methods, both forward and backward transitions, as well as final RMSDs from the target, are reported. Red stop signals indicate methods that were not able to initiate the simulation or provide a transition pathway (see Supplementary Information). Red arrows report the actual computing times for methods requiring more than 12 h. Transition projections in the experimental PC space are also reported. Individual data points used to generate these plots are available in the Source Data file.

undergoes a low-complexity uniform expansion-compression of all protomers (breathing mode), relatively medium-scale (~4 Å RMSD) and highly collective ($\kappa$ ~ 0.9), which can be simulated by eBDIMS2 in just ~30 mins. On the other hand, Nf1 experiences dramatic conformational changes (>23 Å RMSD), but rather localized compared to the overall protein structure ($\kappa$ ~ 0.2), where the GTPase-activating protein-related domain (GRD) and Sec14-PH domain undergo large-scale roto-translations to facilitate Nf1 interaction with its partner Ras[49,50]. As a result, this much more challenging transition can require up to several hours to compute (Fig. 2c; Supplementary Table 7).

It is well known that ENMs are particularly effective at capturing global opening-closing motions[14], a strength intrinsically leveraged by eBDIMS2 to simulate large-scale transitions like the opening-closing movements in the GroEL chaperonin (~15 Å RMSD; Fig. 2d and Supplementary Movie 1), DNA-PKcs (~5 Å; Fig. 5 and Supplementary Movie 2), *SARS-CoV-2* spike glycoprotein (~9 Å; Fig. 6 and Supplementary Movie 3), α-macroglobulin (A2M, ~29 Å RMSD; Fig. 3a and Supplementary Movie 4), etc. Beyond these classical opening-closing motions, eBDIMS2 has also proven effective in handling more topologically complex transitions in high-molecular-weight systems. Notable examples include the full rotary motions of ATP synthases (~14 Å; Fig. 4 and Supplementary Movie 5), the above-mentioned GRD-Sec14-PH roto-translations in Nf1 (~23 Å; Fig. 7b and Supplementary Movie 6), the astonishing ~57 Å rearrangement in the LRP2 dimer (Fig. 3b and Supplementary Movie 7), the ~18 Å shear transition in the ring-shaped macrophage-expressed gene 1 protein (Mpeg1, Fig. 3c and Supplementary Movie 8), the ~18 Å torsional twist in the megadalton-sized Lambda tail tip complex (Fig. 3d and Supplementary Movie 9), etc. These cases underscore eBDIMS2's broad applicability and robustness across virtually all ranges of protein sizes, architectures, and conformational motions. This capability likely arises from the successful combination of a flexible BD simulation framework associated with a pairwise-distance-based biasing approach, instead of strategies based on normal mode analysis (NMA) and RMSD minimization, that are traditionally employed by other CG-methods (see below).

## eBDIMS2 simulates mechanistically consistent pathways for challenging transitions producing intermediates suitable for atomistic applications

Complex transitions not only generally require longer simulation time (see Nf1 and ATP synthases outliers in Fig. 2c), but they are also known to strain simulating algorithms, especially at intermediate high-energy points, generating stereochemical distortions[51]. To assess the overall stereochemical quality of CG models and the ability to sample transitions free of major structural distortions, we computed distances between consecutive $C^\alpha$ atoms for all 191 eBDIMS2 mid-point intermediates in our protein ensemble dataset and compared them to the 124 experimental end-states and all 872 ensemble PDB models. In all cases, distance distributions are centered at the known theoretical value of ~3.8 (Supplementary Fig. 18), with the eBDIMS2 distribution displaying a slightly larger standard deviation compared to experimental structures due to CG relaxation of the backbone. Still, outliers

in the eBDIMS2 distribution are consistent with those already present from experimental models, indicating the absence of major structural distortions induced by the model.

We also compared $C^\alpha$-$C^\alpha$ distances in the intermediates generated by different path-sampling methods for full-length proteins of increasing size. MinActionPath2 is routinely found to induce major structural distortions (with neighboring $C^\alpha$-$C^\alpha$ distances ranging from ~0.2 Å to >30 Å, far from the average 3.8 Å), compromising backbone geometry and rendering intermediates not suitable for atomistic applications. Other methods, including NOLB, aANM, and GOdMD, also show significant backbone deviations in specific systems, while Climber and iMOD are both consistently able to yield near-native CG geometries. While not perfect, eBDIMS2 intermediates are still able to maintain quite realistic $C^\alpha$-$C^\alpha$ distances, with sub-Å deviations from the theoretical values (~3.3-4.3 Å), and avoid the severe artifacts observed in other path-sampling approaches (Supplementary Fig. 17).

We examined in more detail three large systems (~500 kDa) that undergo complex, large-scale conformational changes – *M. Smegmatis* ATP synthase, Nf1 isoform 2, and A2M – two of which appear as outliers in our size–time benchmark (Fig. 2c). We compared the intermediates produced by eBDIMS2 with those from NOLB and MinActionPath2, the only path-sampling methods currently able to handle systems of this scale. Although both NOLB and MinActionPath2 are faster (<30 min) than eBDIMS2, NOLB fails to converge accurately (>10 Å RMSD; Supplementary Figs. 19–21), and both methods generate intermediates that exhibit major distortions in the backbone stereochemistry, with extreme $C^\alpha$-$C^\alpha$ distances ranging from ~2.3 Å to ~18 Å for NOLB, and from ~0.3 Å to ~34 Å for MinActionPath2 (Fig. 4a). In contrast, eBDIMS2 intermediates maintain $C^\alpha$-$C^\alpha$ distance distributions within sub-Å deviations from the theoretical 3.8 Å, and with extreme values close to the outliers already present in the input PDB files (Fig. 4a), thus producing more realistic structures at the CG level.

The rotary motions of ATP synthases are a clear example of exceptional transition complexity. All ATP synthases in our benchmark dataset undergo similar conformational changes, where the rigid rotation of the $F_O$ rotor is coupled with the opening-closing motions of $\alpha$- and $\beta$-subunits during ATP synthesis. By applying rigid constraints to all chains of the $F_O$ rotor (Supplementary Fig. 5), eBDIMS2 is able to guarantee the rotor's rigidity and generate intermediates along the whole synthesis cycles (Fig. 4). The eBDIMS2 transition cycles allow us to recapitulate the experimentally known, alternated, and coordinated opening-closing motions of the three catalytic $\beta$-subunits[52] and in particular how these are coupled to the transmembrane domain rotation (Fig. 4b, c), which was also observed with ns-MD simulations[53].

Finally, to evaluate structural quality at the atomistic level, we also reconstructed intermediates obtained from the different path-sampling methods using cg2all[54] and assessed their stereochemical quality with MolProbity[55]. Consistent with CG-level trends, Climber (atomistic) and iMOD (internal coordinates) always yield the highest-quality structures, while other methods, including eBDIMS and

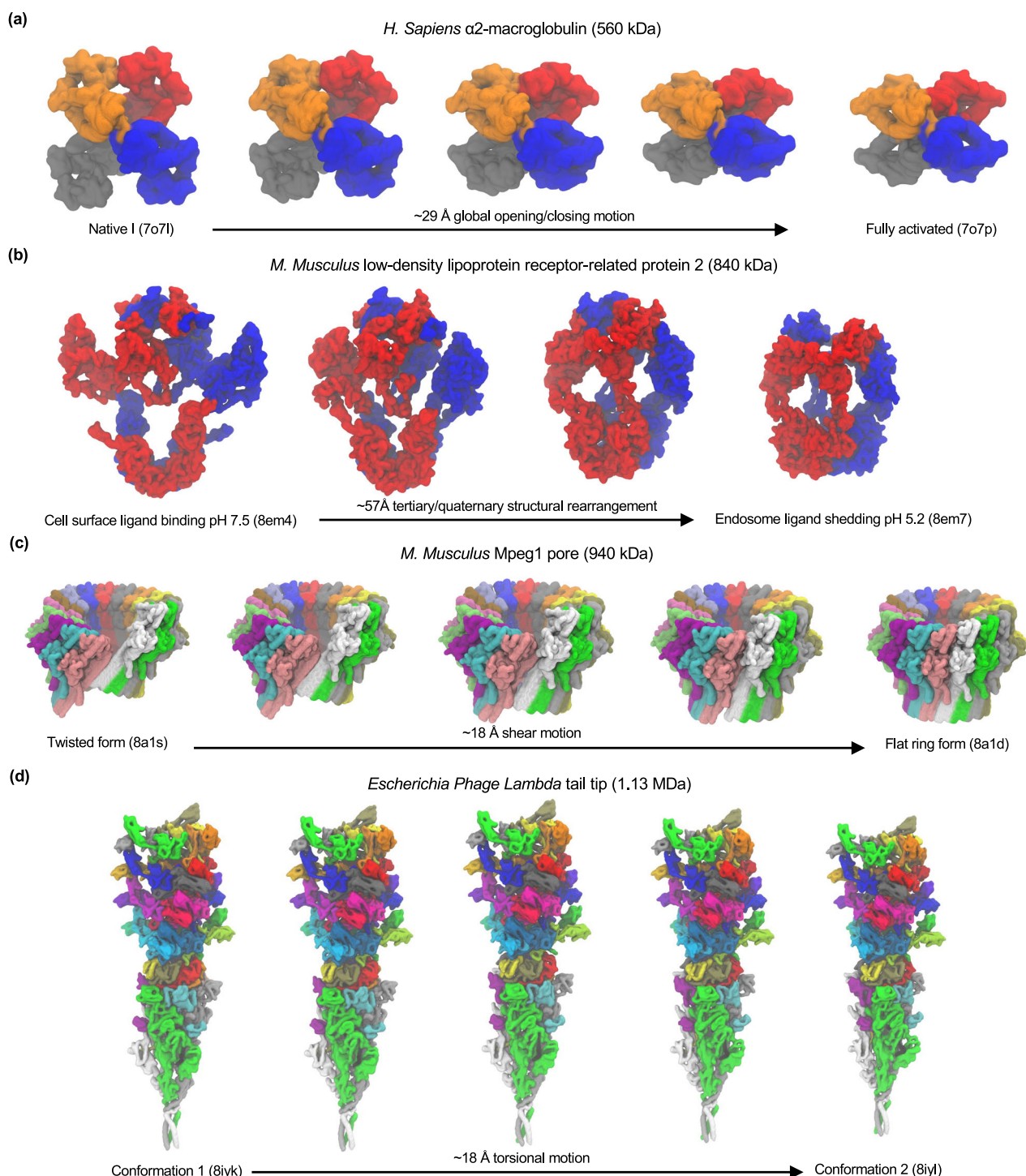

**Fig. 3 | eBDIMS2 has virtually no restrictions on the type/extent of the conformational change and macromolecular architecture. a** Symmetric opening-closing of the four subunits in *H. Sapiens* A2M (~560 kDa) from the native I (PDB: 7o7l) to the fully activated conformation (7o7p). The RMSD of this transition is ~29 Å, with eBDIMS2 converging at ~1.5 Å from the target. **b** Complex rearrangement of the tertiary/quaternary structure in *M. Musculus* LRP2 ( ~ 840 kDa) from a cell surface ligand binding conformation at pH 7.5 (8em4) to an endosome ligand shedding state at pH 5.2 (8em7). The RMSD of this transition reaches an astonishing ~57 Å, with the eBDIMS2 pathway converging at ~4 Å from the target in the closing direction. **c** Shear motion in *M. Musculus* Mpeg1 (~940 kDa) from a twisted conformation (8a1s) to a flat ring state (8a1d). The transition RMSD is ~18 Å, with eBDIMS2 converging at ~1 Å from the target. **d** Torsional motion in *Escherichia Phage Lambda* tail tip ( ~ 1.13 MDa) between two different conformers (8iyk, 8iyl). The transition RMSD is ~18 Å, with eBDIMS2 converging at ~2 Å from the target.

eBDIMS2, initially produce intermediates with significant clash scores and poor MolProbity metrics (Supplementary Tables 9–12). This mostly arises because of suboptimal CG quality, but also due to improper side-chain placements during the atomistic reconstruction. As a matter of fact, short molecular refinement protocols quickly resolve these issues, eliminating structural clashes and improving MolProbity scores to values that are in line with the experimental models (Supplementary Tables 13–16), while maintaining the overall backbone conformation generated by the CG method. These results confirm that, although CG-to-atomistic reconstructions from eBDIMS2

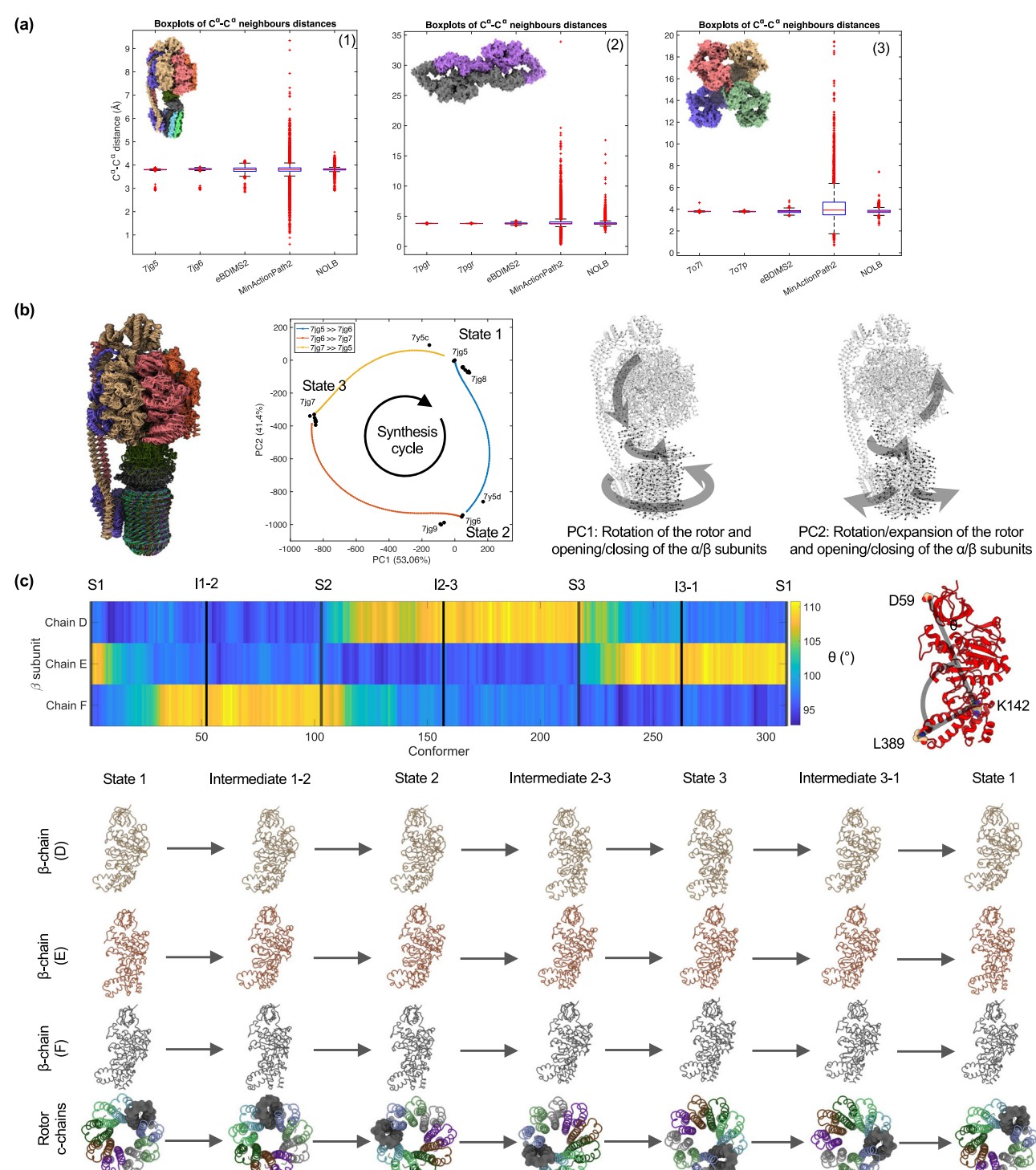

**Fig. 4 | eBDIMS2 simulates mechanistically smooth transitions also for complex and large-scale conformational changes. a** Comparison between the quality of experimental end states, eBDIMS2 and MinActionPath2 transition points, and NOLB final frames for: (1) *M. Smegmatis* ATP synthase, from rotary state 1 (PDB: 7jg5) to 2 (7jg6); (2) Nf1 isoform 2, from active (7pgt) to inactive state (7pgr); (3) A2M, from native (7o7l) to activated state (7o7p). Distances between consecutive $C^\alpha$ atoms are reported using boxplot representations, considering $n = 4684$ consecutive $C^\alpha$-$C^\alpha$ distances for ATP synthase, $n = 4,830$ for Nf1, and $n = 5,074$ for A2M. The bottom and top of each box are the 25th and 75th percentiles of the distance distributions, and the middle red line represents the median value. Whiskers extend up to the minimum and maximum values, while observations beyond the whiskers (red "+" signs) are considered outliers. **b** Synthesis cycle of *M. Smegmatis* ATP synthase simulated with eBDIMS2, generated by merging three trajectories: from state 1 (7jg5) to state 2 (7jg6), from 2 to 3 (7jg7), and from 3 back to 1. A cartoon representation of the cycle is provided (left) together with PC projections (middle). Graphical representations with arrows highlight PC1 and PC2 motions from the experimental ensemble (right). **c** Opening-closing motions of the three $\beta$-subunits (chains D, E, F) during the synthesis cycle, observed from the evolution of the angle $\theta$ between D59, K142, and L389 (right). Angle values are plotted along the synthesis transitions (left) so that dark blue spots ($\theta \sim 90°$) correspond to closed $\beta$-subunit conformations, while bright yellow ($\theta \sim 110°$) spots correspond to open states. Snapshots of $\beta$-subunit motions and *c*-chains of the $F_O$ rotor are also reported (seen from the cytoplasmic side) for each end- and intermediate-state (bottom panel), showing the coordinated opening-closing motions of $\beta$-subunits and their coupling with the transmembrane domain rotation. Individual data points used to generate these plots are available in the Source Data file.

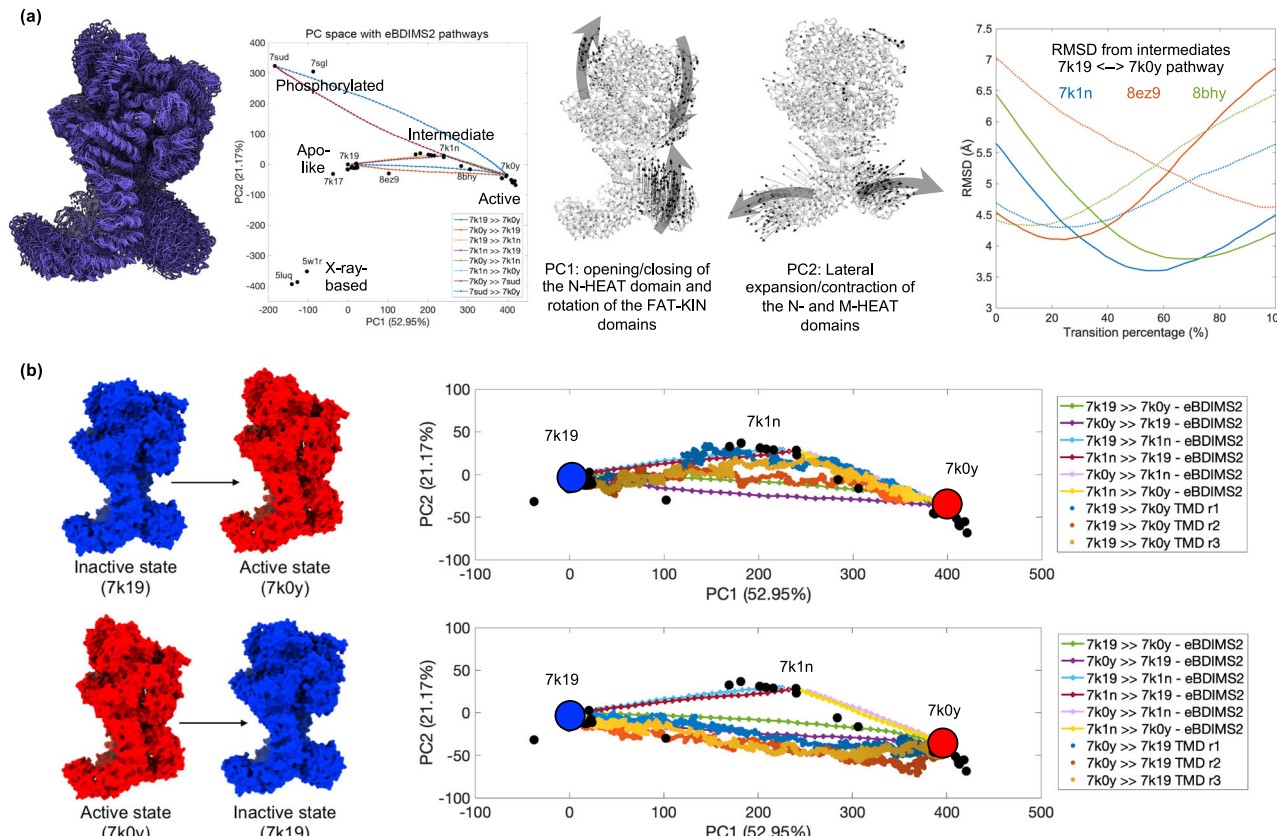

**Fig. 5 | eBDIMS2 pathways capture experimental intermediates and agree with targeted Molecular Dynamics (TMD) simulations: the case of DNA-PKcs.**
**a** Cartoon representation of an ensemble of end-state conformations and eBDIMS2 intermediates of DNA-PKcs (left), together with its projection in the PC space (center), highlighting the cluster of apo-like, intermediate, active, phosphorylated, and X-ray-based conformers. RMSD distance of the eBDIMS2 trajectory between inactive (7k19) and active state (7k0y) from experimental intermediates (right).

Continuous and dashed lines refer to the forward and backward pathway direction, respectively. **b** Comparison between eBDIMS2 pathways and TMD trajectories from the inactive to active state (upper panel) and vice versa (lower panel). eBDIMS2 paths are shown as continuous lines with markers, and the three TMD replicas per direction are represented as points with increasing brightness as the target conformation is approached. Individual data points used to generate these plots are available in the Source Data file.

may be initially suboptimal, the generated intermediates are free from major structural distortions and can readily recover a level of atomistic quality suitable for downstream full-atom applications.

### eBDIMS2 pathways overlap with experimental motions and with enhanced and μs-long MD simulations with minimal computational cost

To further evaluate the biological and mechanistic significance of eBDIMS2 pathways, we assessed whether they spontaneously approach experimental intermediates and MD sampling, as shown previously for smaller systems[26,27]. Identification of potential intermediate states is facilitated by projections of structural ensembles on the low-dimensional spaces defined by the PCs describing the main ensemble motions[26]. Such comparisons go beyond pure stereochemistry, and allow us to evaluate also the smoothness, time-coherence and mechanistic accuracy of trajectories against atomistic data from both trapped experimental intermediates and MD simulation. We selected six systems for deeper study, where intermediate identification is supported by PC projections (Supplementary Figs. 8–12) as well as the literature: DNA-PKcs, the two spike glycoproteins from *SARS-CoV* and *SARS-CoV-2*, ATP-citrate synthase (ACLY), *H. Sapiens* T-complex chaperonin 16-mer (TRiC), and inositol 1,4,5-trisphosphate receptor type 3 (ITPR3). Despite no information on intermediates being fed to the algorithm, eBDIMS2 is always able to approach the existing experimental intermediates with RMSDs as low as ~3-4 Å (Figs. 5a and 6a, right panels; Supplementary Fig. 14), even for these large cryo-EM proteins.

We also performed biased simulations with Targeted MD (TMD)[56,57] for two of the large systems in our dataset, i.e., DNA-PKcs and ACLY, as well as some smaller proteins from our previous benchmark[26]. A significant agreement is generally observed between eBDIMS2 and TMD pathways, which becomes especially evident in cases of marked pathway asymmetries[26], like the opening transition of RNaseIII (Supplementary Fig. 23). An especially interesting case is that of DNA-PKcs (Fig. 5). This large monomeric protein is a fundamental component of the DNA-PK complex, which is central to the process of non-homologous end joining (NHEJ) of DNA[58]. Considering an ensemble of 43 experimental models of monomeric DNA-PKcs (Supplementary Table 4), PC1 is found to cover ~53% of the ensemble variance and corresponds to a vertical motion of the N-HEAT domain, which mediates DNA binding, coupled with a horizontal rotation of the FAT and kinase domains (FAT-KINs). On the other hand, PC2 explains ~21% of the variance and involves a lateral expansion-contraction of the N- and M-HEAT domains (Fig. 5a). Other than a small cluster of X-ray-based conformations, PC projections allow to detect four main functional clusters[58,59]: a cluster of apo-like inactive conformations, where the N-HEAT domain is in the downward position and FAT-KINs are in the inactive inward conformation; a second cluster of intermediate DNA-bound states, where the FATKINs are still in the inactive conformation, but the N-HEAT region has moved upwards into the DNA-binding groove to accommodate DNA-binding; a third cluster of active conformations, where the N-HEAT domain remains in the upward DNA-bound position and the FATKIN head has raised in the active conformation[58], and a fourth cluster of phosphorylated conformers[59].

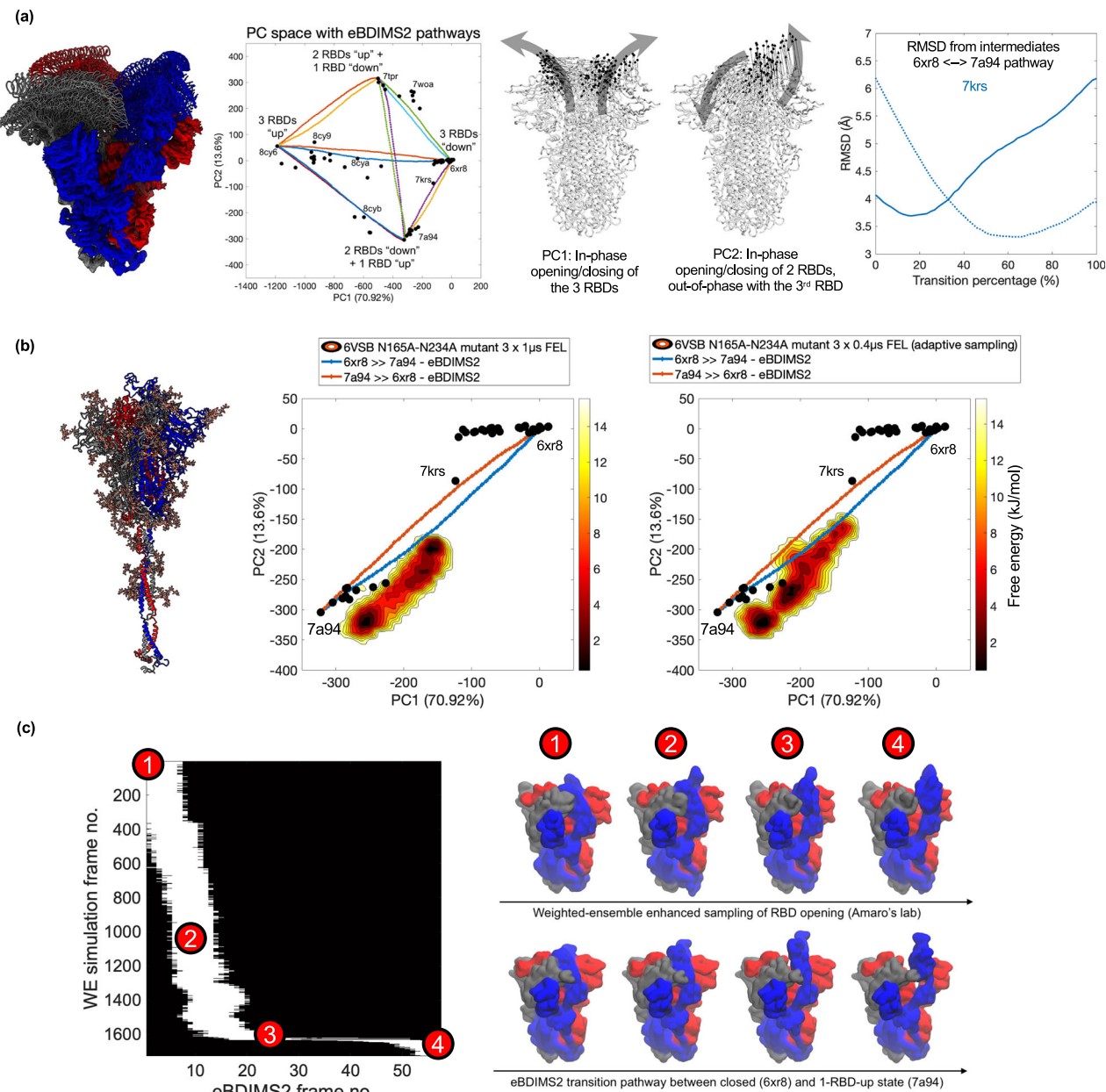

**Fig. 6 | eBDIMS2 pathways capture experimental intermediates and agree with μs-long and enhanced MD simulations: the case of *SARS-CoV-2* spike glycoprotein. a** Cartoon representation of an ensemble of end-state conformations and eBDIMS2 intermediates of the *SARS-CoV-2* spike (left), together with its projection in the PC space (center), highlighting the four conformational clusters corresponding to different opening/closing states of the RBDs. RMSD distance of the eBDIMS2 trajectory between closed (6xr8) and fully open state (7a94) from an experimental intermediate (7krs, right). Continuous and dashed lines refer to the forward and backward pathway direction, respectively. **b** Comparison between eBDIMS2 transition pathways and unbiased (center) and adaptive-sampling (right) MD simulations from Amaro's lab[42] starting from an open conformation with N165A and N234A mutations and glycosylation (6vsb, left). Three independent μs-long MD trajectories have been merged into a single FEL. **c** RBD full opening motion: comparison between the GPU-based Weighted Ensemble (WE) trajectory from Amaro's lab[42] and the eBDIMS2 pathway between the closed and fully open state. Pairwise RMSD comparison (based on minimal RMSD scores, ~4 Å; more details in the Supplementary Information) and visual comparison of WE and eBDIMS2 intermediates along the transition. Individual data points used to generate these plots are available in the Source Data file.

PC1 correlates with the sequential N-HEAT and FATKIN motions required during activation-deactivation of the protein, while PC2 captures the lateral expansions associated with DNA-PKcs phosphorylation. Based on the cryo-EM data, we infer that DNA-PKcs could switch from the inactive conformation (7k19) to the active state (7k0y), and vice versa, directly or via a two-step process visiting the intermediate conformer (7k1n)[58,59]. We simulated several transitions between these end states with eBDIMS2 and verified that the experimental intermediate is approached very closely, even when we simulate a one-step pathway between the inactive and active state (Fig. 5a,

right panel). Moreover, Fig. 5b shows that eBDIMS2 pathways agree with atomistic TMD simulations, as both sample the same area of the conformational space. For the activation, TMD implies that both a one- and two-step mechanism is possible (Fig. 5b, upper panel), whereas for the inactivation mechanism, all simulations suggest a one-step transition without visiting the intermediate conformation (Fig. 5b, lower panel). While describing similar conformational trajectories, the advantage of eBDIMS2 over TMD is that the former does not require a lengthy preparation of the molecular system, and it consumes significantly lower computational resources (~30 min on a desktop

computer with 16 OpenMP threads for eBDIMS2 vs. ~11 h on a high-performance computing cluster with 128 parallel cores for TMD).

We also carried out unbiased MD from end-state and eBDIMS2-generated conformations to explore whether they spontaneously sample the transition (Supplementary Table 17). While some end-state conformers are stuck in low-energy minima, others lead to a broad sampling of the conformational space in agreement with experimental PCs and eBDIMS2 trajectories (Supplementary Table 18). For example, in all simulations from the apo-open conformation of ACLY, this large protein exhibits collective motions along PC1 (-0.8 overlap), with large-scale twisting of the four acetyl-CoA synthetase homology domains, in agreement with the eBDIMS2 paths from an inhibited conformation to a partially open state (Supplementary Figs. 26 and 27). On the other hand, MD simulations from the holo conformation in the absence of ligands show opening motions along PC2, highlighting a spontaneous tendency to go back to the apo-open state. These trajectories are again consistent with the eBDIMS2 holo-apo pathway and capture experimental intermediates that are also closely approached by the eBDIMS2 path (Supplementary Figs. 26 and 27).

Lastly, we compared eBDIMS2 to MD trajectories of the *SARS-CoV-2* spike glycoprotein (Fig. 6a), publicly available from Amaro's lab[42,60]. In this case, unbiased simulations[60] provide only a minor amount of sampling of the spike receptor-binding domain (RBD) motions compared with eBDIMS2 transitions between the experimental end states (Supplementary Fig. 28). Yet, μs-long simulations of a double-mutant (N165A-N234A) spike, which was experimentally shown to reduce binding to angiotensin-converting enzyme 2 (ACE2) receptor as a result of the RBD conformational shift toward the down state[60], were found to provide a FEL consistent with the direction of RBD opening-closing pathways predicted by eBDIMS2 (Fig. 6b). We also compared a Weighted Ensemble (WE) enhanced sampling trajectory, where one RBD was observed to undergo complete opening[42], to the eBDIMS2 pathway from the closed spike (6xr8) to the one-RBD-up conformation (7a94). From the comparison, we found that the two trajectories sample a similar area of the conformational space, showing a good similarity between the WE and eBDIMS2 intermediates (Fig. 6c). Despite employing fundamentally different representations and trajectory sampling strategies, both WE and eBDIMS2 can therefore simulate the RBD opening of the spike glycoprotein. Notably, eBDIMS2 achieves this with minimal system preparation and significantly lower computational cost due to its CG nature, albeit with reduced atomistic accuracy and not being able to address the energetics and kinetics behind the transition. These minimalistic trajectories can provide a fast initial way to explore the conformational landscape for subsequent swarms of MD runs, recovering both energy and kinetics through e.g. Markov State Models[29,30].

## Discussion

Here, we introduce eBDIMS2, an enhanced version of our previous path-sampling ENM-driven Brownian Dynamics (BD) algorithm[26], which achieves over 6-fold speed improvement for large protein systems. This advance enables the CG-simulation of transition pathways in sub-mesoscopic assemblies that were previously infeasible, while retaining its ability to generate mechanistically relevant pathways as shown by our extensive comparisons with experimentally trapped intermediate states and atomistic simulations. The approach is validated through an expanded benchmark of large-scale protein transitions and comparisons with multiple MD techniques, demonstrating its speed, stability, versatility and mechanistic accuracy. To test eBDIMS2, we extended our previous benchmark to include more than 60 large and conformationally diverse proteins ranging from ~300 kDa to ~2 MDa (mostly from cryo-EM, Fig. 2). These proteins undergo transitions from simple ~4 Å breathing motions to highly complex ~15/30 Å rotations-translations, such as Nf1 activation/deactivation pathways and ATP synthase rotatory motions, as well as major structural

rearrangements, such as in the ~60 Å transition of LRP2. Many of the proteins in our benchmark have recently been found to play key roles in several diseases such as cancer[49,50,61–63], tuberculosis[64], skeletal muscle disorders[65], etc. Yet, they have been largely overlooked in molecular simulations due to their extreme size. Using projections on PC1 and PC2 (>70% variance) modes from conformationally rich experimental ensembles, we have identified relevant end states and simulated more than 200 transition pathways, capturing intermediate conformations consistent with our previous benchmark[26].

As observed previously, eBDIMS2 performs comparably to existing path-sampling methods for small- to medium-sized proteins (<1000 residues) while avoiding the limitations of NMA-driven approaches, which may produce abrupt jumps in conformational space despite achieving higher stereochemical accuracy. Both eBDIMS2 and Climber yield smooth transitions along the experimental principal components (PCs), reproducing the sequence of known intermediates and broadly delineating the regions explored by MD. Although slower, Climber maintains perfect stereochemistry owing to its full-atom molecular mechanics force field (ENCAD), making it the best option for smaller systems. The advantages of eBDIMS2 become evident, however, for proteins larger than ~300 kDa, particularly in cases involving highly complex transitions. For example, eBDIMS2 simulates the ~400 kDa GroEL chaperone (~15 Å transition) in ~1 h, whereas the majority of other path-sampling methods fail to complete the task in less than 12 h and often displaying heavy changes in sampling direction. NOLB and MinActionPath2 are the only two other methods currently able to simulate such transition scales, even faster than eBDIMS2, but at the expense of instability (i.e., jumps in sampling) and low target convergence (Fig. 2d) or major structural distortions (Fig. 4a). Across the large ensembles investigated here, eBDIMS2 has a median runtime of ~2.5 h, preponderantly reaching sub-1 Å convergence to the targets while following smooth and mechanistically continuous pathways upon PC-projection. Its computing times scale quasi-linearly with system size (Fig. 2c), contrasting with near-quadratic scaling of the majority of other path-sampling methods.

In MD simulations, $O(N^2)$ complexity is reduced to $O(N\log N)$ with Ewald electrostatics, yet long-timescale transitions remain computationally prohibitive, partly because of the sampling problem. For instance, unbiased MD struggles to capture the *SARS-CoV-2* spike glycoprotein opening[42,60], requiring WE enhanced sampling, which needs to employ several μs of simulation and several GPUs[42]. In contrast, eBDIMS2 can achieve a relatively similar motion description (Fig. 6) in just ~1.2 h on a standard desktop computed using a few CPUs, at the expense of ignoring atomistic interactions and system energetics due to its CG nature. Likewise, for the activation transition of DNA-PKcs, eBDIMS2 reduces computing times from 11 HPC hours to just ~30 min on a desktop, while maintaining overall agreement with Targeted MD (TMD, Fig. 5). Notably, TMD is found to further corroborate the asymmetry of forward and reverse transition pathways, as we previously demonstrated for smaller proteins[26].

Despite the inevitable loss of atomistic detail due to coarse-graining, eBDIMS2-generated intermediates retain sufficient structural quality, even for complex transitions such as the rotary motion of ATP synthase, recapitulating the known, alternated and coordinated opening-closing motions of the three catalytic β-subunits[52] and their coupling to the transmembrane rotation as partially seen in MD[53]. This can enable successful atomistic reconstructions and, upon short molecular refinements, additional downstream atomistic simulations that can help reconstruct complete free energy landscapes[29,30]. In contrast, methods based on NMA (e.g., NOLB) or action minimization (e.g., MinActionPath2) often struggle to define smooth motion coordinates for describing such complex large-scale transitions, frequently introducing major backbone distortions (Fig. 4a). Leveraging a powerful combination between the flexible BD simulation framework and the biasing minimization of internal distances, eBDIMS2 can

accommodate a broad range of conformational changes (Fig. 3) while avoiding major structural artifacts. We foresee that further optimization the ENM force-field with a focus on improving the stereochemical quality of the CG conformers arising from the BD simulation would probably allow us to generate higher-quality models, reducing—or potentially eliminating—the need for subsequent molecular refinement steps.

In recent years, machine learning (ML) approaches such as AlphaFold (AF)[66] and RoseTTA fold[67] have revolutionized the field of protein structure prediction. Beyond static determination, several deep and transfer learning techniques have been developed to address complex biophysical questions, for example, identifying dominant conformational flexibilities from cryo-EM ensembles[68], generating multiple conformations of intrinsically disordered proteins (IDPs)[69] or folded proteins[70] from simulations, and predicting principal directions of protein motion[71]. Recently, several deep learning (DL) methods have been proposed to generate transition pathways between distinct protein conformations by learning from unbiased MD trajectories initiated from the end states[72–74]. A key limitation of these approaches is their dependence on end-state MD sampling, which not only inherits the intrinsic constraints of MD but also restricts their applicability to large (>300 kDa) multimeric systems.

To complement our analyses, we also examined two recent DL-based methods capable of generating time-agnostic ensembles directly from a sequence: BioEmu[75] and AF_unmasked[76]. BioEmu has been trained on extensive databases of static conformations, >200 ms of MD trajectories, experimental stability data, and is currently limited to monomeric and relatively small systems. As a result, it was successfully able to generate ensembles for the smaller proteins in our benchmark dataset, such as RBP and SERCA, but it failed for larger monomeric systems like DNA-PKcs (~450 kDa). The model was found to capture some of the known conformations from X-ray experiments, yet the resulting ensembles tend to display a strong bias toward closed states and include partially unfolded models unsupported by available experimental and MD data (see Supplementary Information). As shown in Fig. 7a, BioEmu has difficulty in reproducing the full range of known experimental conformers of SERCA. The generated ensemble clearly displays a strong bias toward the (closed) E2 state (RMSD < 3 Å). By contrast, the open E1-2Ca²⁺ state and the closed E1-2Ca²⁺-P state are hardly captured, with minimum RMSDs exceeding ~7.5 Å and ~5.4 Å, respectively. The model was also found to generate SERCA conformers exhibiting unfolding of the transmembrane domain and large-scale detachment of the A-headpiece domain, which do not correspond to any experimental evidence currently available. AF_unmasked, by contrast, can be applied to larger and multimeric assemblies. For Nf1, AF_unmasked was found to produce a wide range of 3D conformations characterized by variable scaffold curvatures and GRD−Sec14−PH opening−closing positions[76] (Fig. 7b; Supplementary Fig. 31), but unsupported by currently available experimental data. AF_unmasked can recover the inactive Nf1 conformation, but it fails to capture the active dimer, defined by one monomer in a closed conformation and the other in an open state[50]. Instead, it generates numerous conformers that deviate substantially from the PC-sampled space between the two known end states. The ensembles generated by AF_unmasked (or BioEmu) are not time-correlated, precluding mechanistic insight into inferring the order in which the predicted conformations are visited.

Taken together, these results highlight a fundamental distinction between physics-based path-sampling and current data-driven approaches to conformational sampling (Fig. 7). DL methods generate highly diverse structural models, which can encompass both physically plausible and unfolded conformations unlikely under physiological conditions, and they tend to inherit conformational biases present in the training data (e.g., the overrepresentation of closed states in the PDB, or their more restricted sampling in MD). In contrast,

physics-based approaches such as eBDIMS2 generate reduced yet focused sampling between known experimental conformations, producing time-informed and sequentially ordered intermediates potentially providing mechanistic insight[26,34–36]. We anticipate that a synergistic integration of these complementary approaches will facilitate the exploration of conformational landscapes in complex biomolecular systems.

With minimal preparation – requiring only two sets of PDB coordinates – eBDIMS2 is particularly valuable for the cryo-EM community, which increasingly generates high-resolution structural data[37], but lacks efficient and easy-to-use methods for further mechanistic analysis. By leveraging just a few experimental end-state conformers, cryo-EM researchers can use eBDIMS2 on a standard computer to bridge the gaps in conformational space at the CG-level, then if needed, refine intermediates through atomistic reconstruction and further MD[29,30] (Fig. 1). This capability can make eBDIMS2 a powerful tool for applications ranging from drug discovery and design of binding partners and small molecules[77], to elucidating their mechanism of action[6,78]. To our knowledge, eBDIMS2 is the only algorithm currently capable of generating, with minimal preparation and computational cost, mechanistically relevant and convergent pathways for systems of large size and conformational complexity, with suitable intermediate stereochemistry and in agreement with trapped intermediates and MD, and thus it can accelerate the dynamical and biological interpretation of rapidly growing large-scale structural data.

## Methods

### Protein datasets
In this work, we built a comprehensive dataset of protein ensembles (Supplementary Tables 1–5), with a wide variety of functions, sizes, and shapes (Fig. 2a), including three smaller systems from our previous benchmark[26], i.e., ribose-binding protein (RBP, 30 kDa), RNA endonuclease III (RNaseIII, 48 kDa), sarcoplasmic/endoplasmic reticulum Ca²⁺ ATPase1 (SERCA, 109 kDa). These medium-sized proteins and their structural ensembles of X-ray conformations were used as relevant test cases to assess the ability of eBDIMS2 to replicate our previous results[26], test the performance of the algorithm for different parameters, compare against other path-sampling algorithms, and evaluate the capability of Molecular Dynamics (MD) simulations to explore their conformational space. To retrieve structural data specifically for large proteins ( > 300 kDa) that undergo large-scale conformational changes (RMSD > 4 Å), we performed a far-reaching bioinformatic search from the PDB[38] and UniProt[79] databases. This effort resulted in the construction of high-quality and rich structural ensembles for 47 large proteins, spanning molecular weights from ~300 kDa to ~2.3 MDa (Supplementary Table 4). We also curated an additional dataset comprising 15 two-state protein complexes that undergo large-scale motions (Supplementary Table 5), providing a complementary resource for analyzing the conformational dynamics of large macromolecular assemblies. Detailed information about the screening protocols and the generation of protein datasets is provided in the Supplementary Information.

### Structural ensembles and Principal Component Analysis (PCA)
For all proteins in our ensemble dataset, we generated structural ensembles based on experimental conformations available in the PDB. We retrieved all PDB models based on UniProt ID codes, and we considered all oligomeric states relevant for the protein's biological function (see more details in the Supplementary Information). Structures with low resolution (>5-6 Å) and/or large missing domains were excluded from further analysis. The final list of all PDB models used for the ensemble generation is reported in Supplementary Table 4. We made sure that all the structures belonging to the same ensemble are consistent for further quantitative analyses. We checked that the different chains in multi-chain proteins correspond to the same

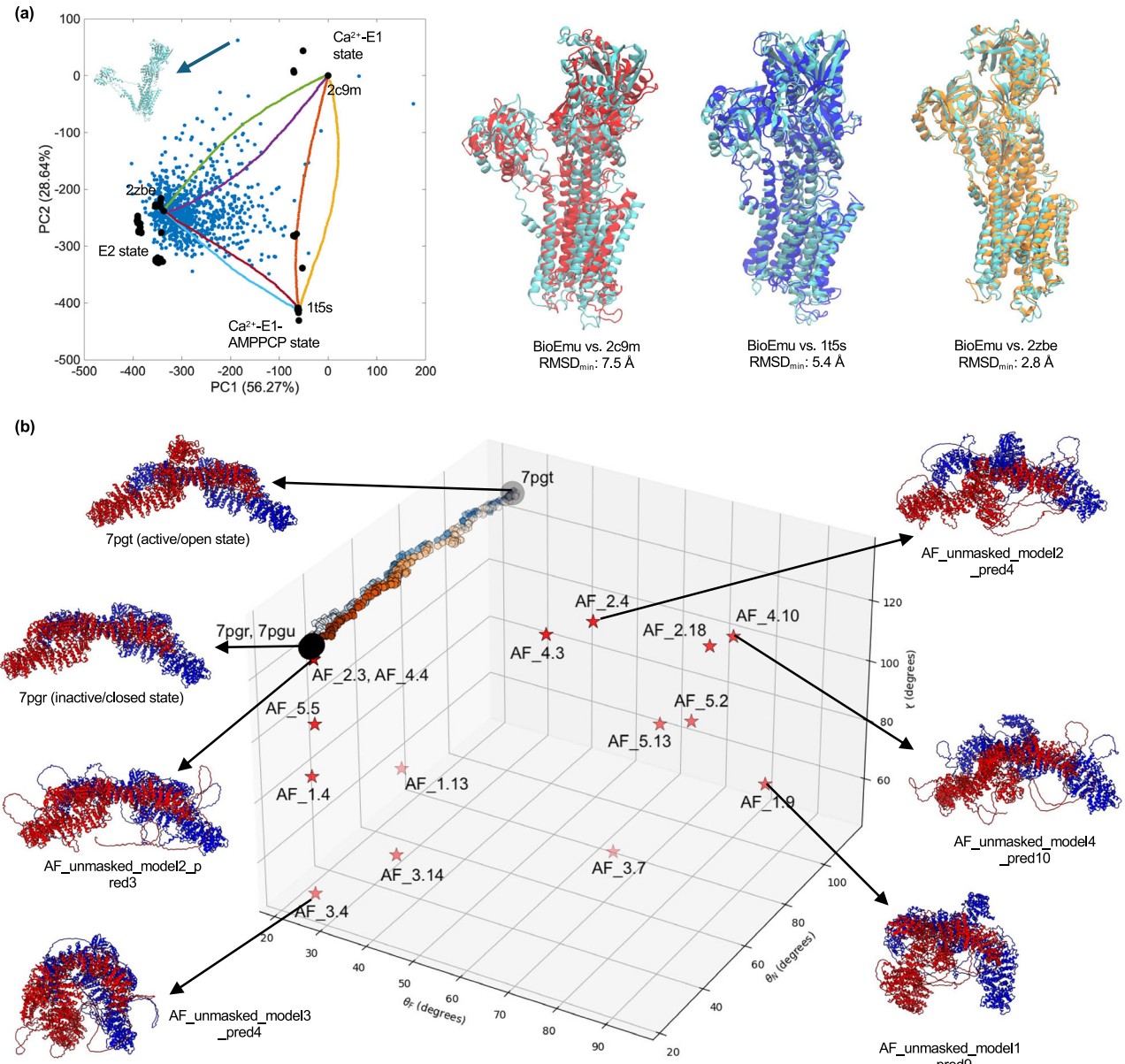

**Fig. 7 | Fundamental differences between DL data-driven methods for ensemble generation and CG physics-based path-sampling approaches like eBDIMS2.**
**a** Comparison between experimental conformations, BioEmu[75] ensemble, and eBDIMS2 transition pathways for SERCA (~110 kDa). The left panel shows the projection of BioEmu predictions (blue points) on the PC space of experimental conformers (black dots), together with the eBDIMS2 pathway between the E1-2Ca²⁺, E1-2Ca²⁺-AMPPCP, and E2 states. The right panel shows the BioEmu conformations (cyan) that are closest to the open E1-2Ca²⁺ (2c9m, red), closed E1-2Ca²⁺-AMPPCP (1t5s, blue), and closed E2 (2zbe, orange) state from X-ray experiments. Note how BioEmu has a clear bias towards the closed (E2) state, hardly capturing the two clusters of E1 conformations (RMSD > 5 Å). The left panel also highlights an example of unrealistic BioEmu conformation with partial unfolding of the transmembrane domain, which leads to complete detachment of the A-headpiece domain.
**b** Comparison between experimental conformations, AF_unmasked[76] ensemble,

and eBIMS2 transitions for Nf1 (~530 kDa). All conformers are projected in a 3D space defined by the average scaffold curvature, χ, and the two angles describing the opening-closing motions of the GRD–Sec-14PH domains of the two Nf1 chains, $\theta_F$ and $\theta_N$. χ and θ angles are defined in the Supplementary Information. Experimental structures from cryoEM are projected as big black dots, while AF_unmasked prediction as red stars and eBDIMS2 intermediates between experimental end states as colored hexagons/pentagons of increasing darkness as they approach the target state. Representative AF_unmasked models, as well as the two experimental states, are also reported with ribbon representation close to the 3D plot. Note how eBDIMS2 pathways cover the conformational space in between the experimental end states, while AF_unmasked tends to sample broader areas of the landscape, with the predicted model exhibiting very different conformations from the experimentally resolved structures. Individual data points used to generate these plots are available in the Source Data file.

protomers, and we removed regions that are missing in at least one conformation to guarantee that all structures have the same number of residues. For each ensemble, a reference structure was selected, generally corresponding to the resting-apo state of the protein. Global structural alignment with respect to the reference structure was applied to all conformations, and PCA was performed on the aligned ensemble.

PCA is a multivariate statistical technique applied to reduce the number of dimensions to describe protein structures and dynamics[80]. PCA has been widely used to describe the essential motions of proteins from MD simulations[81] and experimental ensembles[26,48]. The input of PCA is an $n \times 3N$ coordinate matrix, **X**, $n$ being the number of structures in the ensemble and $N$ the number of residues, usually considering only Cᵅ atoms. From **X**, the elements of the symmetric

covariance matrix, **C**, are calculated as:

$$c_{ij} = \left\langle \left(x_i - \langle x_i \rangle\right)\left(x_j - \langle x_j \rangle\right)\right\rangle \qquad (1)$$

where the brackets $\langle \ldots \rangle$ indicate the average over the n structures. Eigenvalue-eigenvector decomposition is then used to diagonalize the covariance matrix as:

$$\mathbf{C} = \mathbf{U}\mathbf{\Delta}\mathbf{U}^{T} \qquad (2)$$

where the diagonal matrix **Δ** contains the eigenvalues of **C**, while the matrix **U** contains its eigenvectors, representing the Principal Components (PCs). Eigenvalues are sorted in descending order and are directly proportional to the variance captured from each corresponding PC. After calculating the PCs, each structure is projected in the low-dimensionality PC space[26]:

$$p_{l,m} = \left(\mathbf{X_l} - \mathbf{X_{ref}}\right) \cdot \frac{\mathbf{PC_m}}{|\mathbf{PC_m}|} \qquad (3)$$

where $p_{l,m}$ is the projection of conformation l along the $m^{th}$ PC, $\mathbf{X_l}$ and $\mathbf{X_{ref}}$ are the vectors containing the 3D coordinates of the $l^{th}$ conformation and the reference structure, respectively, and $\mathbf{PC_m}$ is the vector of the apparent motion captured by the $m^{th}$ PC. As shown in our previous work[26], PCA of structurally rich ensembles allows us to identify clusters of significant conformations and conformational-functional intermediates. In this work, we used PC projections to select relevant end-state conformers of large proteins and simulate their transition pathways with eBDIMS2.

## eBDIMS2: upgrading eBDIMS to handle large proteins and additional features

eBDIMS2 is an optimized version of our previous elastic network-driven Brownian Dynamics IMportance Sampling (eBDIMS) method[26,46], which is now able to deal with very large proteins and complex conformational transitions in much lower computing times. The goal of eBDIMS is to model conformational changes from a starting conformation, $R_O$, to a target state, $R_t$. It uses a coarse-grained (CG) representation of the protein, considering one bead per amino acid ($C^\alpha$ atom), and implements the MD-derived essential-dynamics Elastic Network Model (edENM) force-field[19], where the protein is treated as a network of mass particles connected by elastic springs. A Brownian Dynamics (BD) framework[82] is used to simulate the protein dynamics and to trace physically acceptable trajectories from $R_O$ to $R_t$. The equations of motion follow the Langevin equation[26]:

$$m_i\ddot{r}_i = F_i - \gamma\dot{r}_i + \xi_i \qquad (4)$$

where $m_i$ and $r_i$ are the mass and the position of the $i^{th}$ particle, respectively, $F_i$ is the force acting on the $i^{th}$ particle due to the particle-particle interactions from the edENM, $\gamma$ is the friction coefficient related to dispersion forces arising from the interactions with the surrounding fluid[82], and $\xi_i$ is a time-dependent white-noise term that accounts for the thermal motion of the solvent[9,26,82]. In order to bias the trajectory in the direction of the target, eBDIMS uses Dynamics IMportance Sampling (DIMS). Every number k of BD steps (biasing frequency), a progress variable Γ is computed and used to drive the transition. Γ is defined as the difference in pairwise distances between the simulated ($R_s$) and target ($R_t$) conformation[26]:

$$\Gamma_s = \sum_{i=1}^{N-1}\sum_{j=i+1}^{N}\left(d_{ij}{}^s - d_{ij}{}^t\right)^2 \qquad (5)$$

where $d_{ij}{}^s$ is the distance between the $i^{th}$ and $j^{th}$ particles in the simulated structure $R_s$ at step s, $d_{ij}{}^t$ is their distance in the target

conformation $R_t$, and N is the total number of particles in the system. $\Gamma_s$ is compared every k steps to the previous value, $\Gamma_{s-1}$, and the current conformation $R_s$ is accepted if $\Gamma_s < \Gamma_{s-1}$, or rejected otherwise[26]. The iterations proceed until convergence to $R_t$, e.g., until the sampled conformations reach a Root Mean Square Deviation (RMSD) from the target in the range of thermal oscillations (~1 Å) or when $\Gamma_s$ is sufficiently close to zero.

Our original version of eBDIMS is currently available as a public web server and as a stand-alone C++ code[46] and is efficient for proteins up to ~2k residues (~250 kDa). Larger systems would require an enormous time to drive the transition up to the target conformation. The updated eBDIMS2 algorithm, now implemented in Fortran, addresses the scalability limitations of the previous code by introducing a more efficient force computation strategy during BD simulations. In the original approach, the force on each particle $F_i$ was calculated by summing over all pairwise interactions:

$$F_i^{BD} = \sum_{i=1}^{N-1}\sum_{j=i+1}^{N} F_{ij} \qquad (6)$$

where $F_{ij}$ represents the edENM force between particles i and j. This results in a computational cost that scales quadratically with system size: $(N^2 - N)/2$ interactions for a system of N particles. Even a relatively small protein such as RBP (271 residues) requires ~36,500 pairwise force evaluations per step, while a large system like the ryanodine receptor 1 (RyR1, ~17,000 residues) involves nearly 140 million interactions, rendering the original algorithm impractical for large biomolecular systems. Due to the power-law decay of edENM interactions between non-bonded particles and the adoption of a spatial cutoff in the underlying ENM, eBDIMS2 adopts a more efficient force computation strategy based on the implementation of an adaptive cutoff for the selection of interacting residue pairs in the BD simulation – akin to standard practices in MD. At the start of the BD simulation, we construct a list L of proximal residue pairs in the reference conformation $R_0$, using the ENM cutoff $r_c$ such that all pairs satisfying $d_{i,j} < r_c$ are included, where $d_{i,j}$ is the distance between particles i and j. This list is then used to compute only the non-negligible pairwise forces $F_{ij}$, significantly reducing the number of calculations:

$$F_i^{BD} = \sum_{(i,j)\in L} F_{ij} \qquad (7)$$

This approach dramatically lowers computational cost, particularly for large systems. With a cutoff $r_c = 8$ Å, eBDIMS2 requires ~1.6k interactions for RBP (compared to ~36.5k in the original eBDIMS), and ~80k interactions for RyR1 (instead of ~140 million). To account for conformational changes during the simulation, the interaction list L is periodically updated based on the new positions of the $C^\alpha$ atoms. Then, the simulation proceeds until convergence. We tested the performance of eBDIMS2 for different values of the cutoff $r_c$ (8, 10, 15, 20 Å) and the biasing frequency k (1, 2, 5, 10), looking at the time required to simulate the transitions and their projection in the PC space. While these parameters were not found to play a significant role in the PC projections of the transitions (Supplementary Fig. 4), a cutoff $r_c$ of 8 Å and a biasing frequency k of 10 steps were found to be optimal to minimize the computing time (Supplementary Fig. 3).

Another advantage of eBDIMS2 is that it can now compute pathways between structures with missing residues. Most of the path-sampling methods available in the literature[83] require the two end-state protein conformations not to have missing residues. However, almost all large systems from cryo-EM inevitably present several regions that are missing in the 3D model due to, e.g., difficulties in fitting density maps, low resolution, high local flexibilities, etc. For this reason, we have developed eBDIMS2 in such a way that the two protein end states can have gaps in the sequence (Supplementary Fig. 6).

The updated version of our algorithm also includes two additional features. First, eBDIMS2 can now handle motions involving quasi-rigid domain rotations, where standard ENM approaches often introduce artifacts during large-scale rotations. This is overcome by increasing the spring connectivity (removing the cutoff for long-range interactions) and stiffness constants (all spring constant values set to 1 kcal/mol.Å2) within these rigid regions. In this way, eBDIMS2 can better preserve the domain structural integrity in purely rotational motions (Supplementary Fig. 5). In this work, this feature has been applied to all ATP synthases in our dataset at the whole-chain level, making all residues of the individual rotor chains part of a rigid block. Second, the algorithm can now also support transitions between end states with differing residue counts or chain compositions. Residue correspondence is established through matching chain identifiers and residue numbering, enabling transition convergence even when entire domains or chains are missing in one of the two end-state conformers. In these cases, the common regions drive the transition, while unmatched segments usually undergo free fluctuations, enabling the modeling of conformational changes also for incomplete assemblies (Supplementary Fig. 7). Further details are provided in the Supplementary Information.

## Application of eBDIMS2 to large-scale conformational changes and stereochemistry assessment

After identifying biologically relevant conformational clusters from the PC spaces of our ensemble dataset, we applied eBDIMS2 to run transition pathways between all end-state conformers, both in the forward and reverse directions. Similar simulations were also performed for the additional dataset of two-state proteins, using the more flexible version of the code that allows the two protein end states to have different numbers of residues and/or chains. All calculations were carried out on a Linux workstation with an Intel® Core i9-13900K processor, 64 GB of RAM, and using OpenMP parallelization with 16 threads. For each transition, we computed RMSD values and collectivity degrees. The former quantifies the amplitude of the conformational change:

$$RMSD = \sqrt{\frac{1}{N}\sum_{i=1}^{N}\left(r_i^{\,t} - r_i^{\,0}\right)^2} \qquad (8)$$

where $r_i^{\,t}$ and $r_i^{\,0}$ represent the positions (after global structural alignment) of the common $i^{th}$ $C^\alpha$ atom in the target and reference conformation, respectively. The collectivity degree $\kappa$ provides an estimate of the global-local nature of the transition[14]:

$$\kappa = \frac{1}{N}\exp\left(-\sum_{i=1}^{N}\frac{|r_i^{\,t} - r_i^{\,0}|}{\sum_{i=1}^{N}|r_i^{\,t} - r_i^{\,0}|}\ln\frac{|r_i^{\,t} - r_i^{\,0}|}{\sum_{i=1}^{N}|r_i^{\,t} - r_i^{\,0}|}\right) \qquad (9)$$

and its value can range from a minimum of 1/N, when only one atom is involved in the conformational change, to a maximum of 1 when all atoms uniformly participate to the transition. For the proteins in the ensemble dataset, eBDIMS2 pathways were then projected on the corresponding PC spaces via Eq. (3), to inspect the relationship between the generated intermediates and experimental conformations. To quantify the performance of the method, we recorded RMSD values from the target at the moment of convergence and the time employed by the method to reach convergence. We also computed RMSD values between eBDIMS2 transitions and on-path experimental intermediates.

To assess the stereochemical quality of the intermediates generated by eBDIMS2 at the level of the CG backbone, we computed distances between pairs of consecutive $C^\alpha$ atoms and compared these $C^\alpha$-$C^\alpha$ distance distributions to those obtained from other path-sampling methods and experimental models. Then, to assess the quality of the

generated intermediates also at the atomistic level, we made use of the MolProbity validation tool[55]. CG models first underwent atomistic reconstruction with the recently developed method cg2all[54], and in a few cases, we also carried out molecular refinement protocols encompassing a short energy minimization, equilibration, and a short MD production run (Supplementary Information).

## Comparison with other path-sampling methods

Over the past 20 years, a plethora of methods have been developed to model conformational transitions in proteins[83]. Here we compared eBDIMS2 with our previous eBDIMS $C^{++}$ version[26] and nine additional algorithms whose executables are available: iMOD[20], GOdMD[84], NGENI[85], ICONGENI[86], Climber[27], NOLB[45], ENI[87], aANM[88], and ANMPathway[51]. These methodologies mainly differ for: (i) the representation of the protein degrees of freedom (DOFs); (ii) the simulation framework to model the protein dynamics; (iii) the biasing strategy used to drive the transition; and (iv) the reversibility-irreversibility of the transition in the forward-backward direction. More information about these methods is provided in the Supplementary Information. The majority of these algorithms do not allow to analyze transitions in proteins with missing residues. For this reason, we carried out a detailed comparison for four proteins in our dataset that have full-length structural ensembles of increasing molecular weight, i.e., RBP (271 residues), RNaseIII (432), SERCA (993), and GroEL in the 7-mer oligomerization state (3626). For these four systems, we simulated transition pathways between the two relevant end-state conformations, and we compared computing times, convergences, and pathway projections in the PC space. All calculations were performed on the same Linux workstation described above. We also compared the stereochemistry of the generated intermediates at the level of the CG backbone, as well as for the atomistic models reconstructed with cg2all[54] via MolProbity[55].

Very recently, MinActionPath2 was released[44], which represents an improvement of the previous MinActionPath algorithm[89], and can now be used to deal with large macromolecular assemblies. Both MinActionPath and MinActionPath2 are only available as webservers, which prevents a thorough comparison of computing times with eBDIMS2. While we did not use these tools for the time and PC projection comparison, we used the MinActionPath2 webserver to assess the stereochemical quality of the transition points for the four full-length proteins mentioned above, as well as for three larger proteins in our ensemble dataset that undergo complex and large-scale transitions.

## Atomistic MD simulations

We also performed atomistic MD simulations, both unbiased MD from end-state and eBDIMS2-intermediate conformations to assess the sampling of the conformational landscape, and Targeted MD (TMD) to simulate transition pathways[56,57]. For proteins with missing residues, e.g., DNA-dependent protein kinase catalytic subunit (DNA-PKcs) and ATP-citrate synthase (ACLY), unmodelled gaps were filled by using the SWISS-MODEL[90] webserver. For simulations starting from eBDIMS2-generated intermediates, cg2all[54] was used to obtain atomistic models suitable for MD.

All molecular systems were prepared with CHARMM-GUI[91], and MD simulations were performed using Gromacs[92] version 2024.1. For TMD, we used Gromacs patched with Plumed[93]. The CHARMM36m force field was used to describe the biomolecular interactions[94], and we added TIP3P water molecules as well as sodium (Na$^+$) and chloride (Cl$^-$) ions at 150 mM concentration, to maintain physiological salt concentration and mimic intracellular conditions. First, we carried out an energy minimization with the steepest descent algorithm for 5000 steps. Then, the system underwent a 125-ps equilibration in order to maintain a temperature of 303.15 K, with the Nose–Hoover thermostat[95,96], and a pressure of 1.0 bar, using the

Parrinello–Rahman barostat[97] with isotropic pressure coupling. The LINCS algorithm was used to constrain H-bonds[98]. Short-range van der Waals and electrostatic interactions cutoffs were set to 12 Å. Long-range electrostatic interactions were described using the particle mesh Ewald (PME) approach[99,100] with periodic boundary conditions.

For unbiased MD simulations, we carried out production runs using a 2-fs time step and saving coordinate frames every 100 ps. To speed up the simulations for medium-size proteins, we made use of H-mass repartitioning with a longer 4-fs time step[11]. Each system was simulated for 200 ns, with three replicas starting from different random seeds. A total of 0.6 μs unbiased atomistic dynamics was generated for at least two distinct conformations (Supplementary Table 17), which we used to build Free Energy Landscapes (FELs). To check how well MD trajectories align with experimental conformations, we computed overlap scores and Root Mean Square Inner Products (RMSIPs) between experimental PCs and Essential Dynamics (ED) eigenvectors from MD[81]. For TMD, three production runs were carried out to obtain transition pathways between the selected end-state conformers. For the medium-size systems, TMD runs were carried out for 1 ns, while for DNA-PKcs and ACLY, we simulated for 2 ns (Supplementary Table 17). In TMD, the RMSD between the two aligned end states was used as bias and applied every 10 steps with an elastic constant of 100 kcal/molÅ$^2$. TMD trajectories reached convergence in all analyzed systems, with an RMSD from the target of ~1-2 Å.

Since our benchmark dataset included the much-studied *SARS-CoV-2* spike glycoprotein, we also made use of MD trajectories publicly available from the Amaro's lab[42,60]. We downloaded several simulations of the open, closed, and N165A-N234A double-mutant states of the spike[60], as well as a trajectory showing the opening of one receptor-binding domain (RBD) obtained through a Weighted Ensemble (WE) enhanced sampling approach[42]. All trajectories and FELs were then projected on the experimental PC spaces and compared with our eBDIMS2 pathways.

## Reporting summary

Further information on research design is available in the Nature Portfolio Reporting Summary linked to this article.

## Data availability

The data of ensembles and eBDIMS2 transition pathway, with all related PDB files, generated in this study, have been deposited in the figshare database under accession code 29423201[101]. The starting .gro files used for all unbiased and targeted MD simulations, as well as protein Cα-only trajectories in multi-model .pdb format, have been deposited in the figshare database under accession code 31086934[102]. The source data underlying Figs. 2b–d, 4a–c, 5a, b, 6a–c, 7a, b, and Supplementary Figs. 2, 3, 16, 17 are provided as a Source Data file. Source data are provided with this paper.

## Code availability

The eBDIMS2 code is available at [https://github.com/domenicoscaramozzino/eBDIMS2][103].

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

## Acknowledgements

L.O. acknowledges financial support from Cancerfonden Junior Investigator Award (CF 21 0305 JIA) and Project Grants (CF 21 1471 Pj, CF 24 3801 Pj) as well as Vetenskapsrådet Starting Grant (VR 2021-02248) and Karolinska Institutet. D.S. acknowledges financial support from Cancerfonden postdoctoral fellowship (CF 24 0908 PT). Simulations were run using the National Academic Infrastructure for Supercomputing in Sweden (allocations NAISS 2023/5-400 and 2024/1-7 to L.O.).

## Author contributions

D.S. developed eBDIMS2, built the protein datasets, generated structural ensembles, designed and performed all simulations and analyses, prepared figures, and wrote the manuscript draft. B.H.L. contributed with MD simulations and with comparison with ENI, NGENI and ICON-GENI path-sampling algorithms. LO conceived the original idea and contributed to discussions. All authors participated in data interpretation and manuscript revision.

## Funding

## Competing interests

The authors declare no competing interests.
