## [Transparent Peer Review file · Nature Communications]

Efficient sampling of large-scale transition pathways and intermediate conformations in sub-mesoscopic protein complexes

Corresponding Author: Professor Laura Orellana

Version 0:

Reviewer comments:

Reviewer #1

(Remarks to the Author)

The authors present an update of eBDIMS (Nat. Commun. 2016 & Bioinformatics 2019) and its application to a set of proteins assembled with at least three cryo-EM models. The article is well-written, but I have several major concerns regarding the following aspects:

1. **Originality:** As I understand it, the only methodological differences from the previous version are incorporating a nearest neighbor list to reduce computational cost and including a mask to handle proteins with missing residues. Both are standard procedures in the field, and the novelty of these additions is not adequately highlighted.
2. **Context:** The authors should situate their methodology within the current landscape of conformational ensemble experimental characterization and prediction. Cryo-EM has the potential to characterize the conformational space in solution and directly capture the ensemble conformations. Current and future efforts in the cryo-EM data processing are increasingly incorporating deep learning (DL) approaches (e.g., <https://www.nature.com/articles/s41467-024-49858-x>). Additionally, DL methods aimed at generating physically realistic conformational ensembles from single conformations are gaining traction (including transitions, <https://doi.org/10.1038/s41467-023-36443-x>). How does your approach fit in this context? Related to this, the authors conducted an exhaustive search throughout the entire PDB and only identified 45 test cases, many of which are chaperones. This limited dataset could give the false impression of restricted applicability. The authors should make an additional effort to identify more transitions, such as multimeric complexes, viruses, ribosomes, and other larger assemblies, to better demonstrate the method's utility.
3. **Comparison with Other Methods:** In previous publications, the authors have already compared different methods, and significant differences with larger structures are not necessarily expected. However, the current manuscript should include more recent methods, such as MinActionpath2 (only partially) or NOLB, which can also be applied to large systems (<https://doi.org/10.1016/j.bpj.2020.03.020>). Additionally, DL approaches should at least be acknowledged.
4. **Suspicious Results:** There are some unexpected results that should be clarified. For instance, ANMPPathway is reported to be non-functional for a 432-residue RNAsell but works for twice the size SERCA. Similarly, the classical example of the GroEL 7-mer is reported to be unsuccessful except for ebDIMS2. The default parameter settings and input structures of other methods may require fine-tuning for larger models.

Minor Comments:

- Are other methods capable of running on multiple processors? Why is computational efficiency emphasized so strongly?
- The authors frequently claim “realistic pathways”, yet after regularization with cg2all, eBDIM2 exhibited sub-optimal Ramachandran values. Structural quality assessments have only been performed for one case (GroEL, Table 7). What about RBP, RNAsell, and SERCA? And how do other methods compare in terms of structural quality assessments?

(Remarks on code availability)

Strangely, the code has been translated to Fortran (I am not an expert) from the previous C++ version.

Reviewer #2

(Remarks to the Author)

I recommend publication with minor edits.

In this manuscript Scaramozzino et al. present the eBDIMS2 method, which provides an efficient and capable method of sampling transition pathways between multiple protein conformational states. While the method is impressive and applied to, in my view, enough impressive examples, I believe the presentation of the method, especially in comparison to existing path sampling approaches, could be improved to highlight the nuance in choosing a method for sampling pathways. Below are my specific comments.

- In line 65, I appreciate the comparison between eBDIMS2 and other path sampling approaches, and I agree that eBDIMS2 is much more time efficient. However, it is worth pointing out that for the WE simulations you mentioned, the goal was not just to quickly generate a transition pathway. In WE simulation, one often attempts to simulate multiple (50-100) transition pathways, so the time reported here likely reflects that fact. It would be more apples-to-apples to compare the time it took WE to achieve the first transition path, which I am not sure the authors of the original paper reported. In any case, perhaps a parenthesis to this comparison could be added to explain this nuance.
- Throughout, it was not immediately clear to me what makes eBDIMS2 different from eBDIMS1. I can understand and appreciate the scale of systems that version 2 is now able to simulate, but not what it is about version 2 that allows that. It would be nice for the authors to explain this in more detail in the main text.
- Regarding the use of "surpasses" in line 76 and related language. I am not convinced that eBDIMS/2 "surpasses" all existing path sampling methods, as there are pluses and minuses to each method. I think the wording throughout could be made a little more nuanced so that readers have a better appreciation of the actual strengths and limitations of the method compared to more traditional path sampling strategies.
- In line 136, why are smoother pathways better? Wouldn't that reflect a divergence from the underlying, detailed physical models?
- Figure 2: Why are the forward and reverse paths sampled by eBDIMS2 different? Is this expected? If this has not been addressed in previous eBDIMS papers, maybe this could be commented on here, as I would have expected the forward and reverse paths to appear more similar.
- Figure 3: Were the authors able to generate reverse pathways for the ATPase? If not, what prevented doing so?
- In line 528, how exactly does the eBDIMS2 workflow account for sequence gaps?
- To my view, eBDIMS2 achieves such a large speedup compared to methods like WE from the use of CG models. This could also be viewed as a limitation of the method. How do you interpret a CG pathway in terms of fully atomic structures? What approximations are you making and do you think they are correct to make? Maybe I missed these details, but a discussion on this would, I think, be warranted.

(Remarks on code availability)

I appreciate that the code is available on GitHub. I have reviewed it. It seems to be a straightforward Fortran script. There do not appear to be a lot of installation/usage instructions, though, so non-code-savvy users may struggle. Though I did not try, I should be able to install and run the code.

Version 1:

Reviewer comments:

Reviewer #1

(Remarks to the Author)

The readers should be aware that the algorithmic differences compared to the previous version are relatively straightforward, and the overall algorithmic novelty appears limited. The introduction of a spatial cutoff to reduce the number of particle interactions is a simple and obvious improvement that arguably should have been incorporated in the earlier version. Additionally, the possibility of incorporating rigid blocks within the structure is not clearly explained in the main text. How is this implemented? Is it chain-based? Is this feature optional or applied generally across all simulations?

As previously mentioned, there are some unexpected results that require further clarification. For instance, the classical GroEL 7-mer example, which is reported in Figure 2 as unsuccessful in other methods, has actually shown successful outcomes in several other tools. For example:

-MinActionPath2: <http://www.dynstr.pasteur.fr/servers/jobs/68c7f4988c977998727255/>
-iMODs: https://imods.iqf.csic.es/Results/job_GroELring_CO/index2.html

I suspect that other cases, such as NLOBS, might also yield positive results using alternative methods. It is also well known that NMA-based approaches often struggle with "closed-to-open" conformational transitions, typically performing better in the "open-to-closed" direction. Therefore, the results presented in Figure 2 should be revisited and revised accordingly.

In the case of GROEL, as well as in many others, the initial mapping (or structural alignment) is critical and should be carefully re-evaluated. Additionally, many of the methods compared were not originally designed to handle large molecular complexes, which makes a direct comparison somewhat problematic.

Rather than focusing on a single structure, such as the Nf1 example, it would be more informative to evaluate deep learning (DL) approaches on the same set of cases used to benchmark the other methods. The authors chose to compare with AF_unmasked, but there are other recent and relevant DL-based methods worth considering, for example:

<https://pubs.acs.org/doi/10.1021/acs.jctc.4c00816>
<https://www.science.org/doi/10.1126/science.adv9817>
<https://www.nature.com/articles/s41467-024-55228-4>
<https://pubmed.ncbi.nlm.nih.gov/40060558/>

The claim of producing "realistic pathways" appears somewhat overstated, particularly when accompanied by the statement that "other methods, including eBDIMS and eBDIMS2, initially produce intermediates with significant clash scores and poor MolProbity metrics." It is likely that the poor stereochemistry observed originates from the quality of the coarse-grained models, rather than from the Cg2All regularization step itself.

While additional refinement or regularization steps (e.g., short molecular dynamics or energy minimization) could alleviate these issues, this raises the question: what is the advantage of coarse-grained approaches if high-resolution refinement is ultimately required (disregarding other problems such as missing residues, etc.)? Furthermore, could the authors comment on the relationship between poor MolProbity metrics and the preservation of realistic C α -C α distances in the generated models?

Finally, I recommend tempering expressions such as "realistic pathways" or "high-resolution models," which may not fully align with the evidence presented.

That said, I truly appreciate the authors' effort to expand the range of test cases. In my view, this significantly increases the value of the article. As another reviewer has already suggested, the implementation of a public web server would be an excellent addition to broaden accessibility and usability of the method.

(Remarks on code availability)

I did not run the fortran code, but I checked the benchmark material and its ok (a bit lengthy)

Reviewer #2

(Remarks to the Author)

I appreciate your detailed and thoughtful responses to my comments. As of now, all of my concerns have been addressed. I believe your changes have made the manuscript even stronger.

(Remarks on code availability)

Version 2:

Reviewer comments:

Reviewer #1

(Remarks to the Author)

The authors have satisfactorily addressed the major concerns raised in my previous review. The manuscript has improved substantially, and only minor comments remain:

1.1

Readers should note that the algorithmic differences relative to the previous version are relatively straightforward. As the authors themselves acknowledge, although practically effective, the modifications are algorithmically simple. The principal factor enabling the extension to large model transition, the introduction of a distance cutoff, has been a standard feature in coarse-grained models since the early stages of ENM development (Phys. Rev. Lett. 1996, 77:1905–1908). From a methodological perspective, the novelty is limited; nonetheless, the main value of this contribution lies in the successful application of eBDIMS2 to large macromolecular complexes.

1.2

The explanation of the rigid-block pre-definition is now substantially clearer and satisfactory.

1.3–1.5

In my previous review, I was under the impression from Figure 2 (and Supplementary Figure 16S) that none of the methods except eBDIMS2 could model GroEL. I now understand that this misunderstanding arose from the arbitrary 12-hour timeout threshold. To avoid misinterpretations and ensure fair comparison, I recommend removing timeouts entirely so that readers can better assess both the actual relative efficiency of the methods and their proximity to the target conformation.

Additionally, in panel 2d, truncated and complete pathways appear to be mixed. In Supplementary Figure 15S, the open–close trend characteristic of NMA-based methods should also be visible for GroEL, as mentioned in response 1.5.

From the outset, I expressed concerns about potential pitfalls in the initial structural alignment and the intrinsic difficulty of

such comparisons. I appreciate that the authors identified the alignment error in MinActionPath2. Since most path-sampling methods are RMSD-based and depend on an initial structural alignment (typically local, aligning the most rigid regions), the use of unaligned inputs has limited interpretative value. Thus, the rationale for including “unaligned” runs in Table R1 remains unclear. I assume that these correspond to a different local or global alignment algorithm. It is evident that all methods are sensitive to initial conditions, and studying this dependence in detail would be complex. My earlier suggestion merely aimed to encourage careful verification of potential alignment errors.

1.6

The principal value of the present contribution lies in demonstrating that eBDIMS2 can now handle megadalton-scale assemblies. It is evident that existing path-sampling methods are not readily applicable to such large systems. Nevertheless, readers may be more interested in the practical applications than in extended comparisons with methods not designed for large molecular assemblies. It should also be acknowledged that some of those methods could be made more efficient or achieve lower RMSDs by tuning their parameters for specific cases—for example, by modifying the elastic network, employing parallelization, using rigid-block pre-definition, or increasing the level of coarse-graining.

As an illustration, the classical RTB-NMA approach has been successfully applied to model large-scale conformational transitions between distinct structural states of viral capsid proteins

(<https://www.sciencedirect.com/science/article/pii/S0022283602001353>).

1.7

The comparative results for RBP and SERCA are particularly noteworthy and, in my view, deserve greater prominence in the main text. Given the increasing popularity of AI-based approaches, many potential users are likely to consider BioEmu as their first option. In this context, Supplementary Figure 30 provides an excellent and important cautionary example.

Considering the manuscript's length limitations, I strongly recommend replacing one of the existing figures with Figure 30.

1.8 & 1.10

I was intrigued by the potential correlation between MolProbity scores and C α –C α distances, and I appreciate the authors' clarification regarding C α -RMSD. I do not dispute that C α –C α distances primarily reflect backbone integrity. However, given that the standard deviation of the trans peptide bond from the average C α –C α distance (3.8 Å for trans) is approximately 0.04 Å, deviations greater than five standard deviations may indicate possible distortions. In any case, I concur that the cg2all tool is a useful addition that can correct such distortions in the backbone and further minimization also remove side-chain collisions.

1.9

In general, the limited structural quality of large assemblies, the size and the presence of missing residues make atomistic studies difficult or even infeasible. In this regard, eBDIMS2 represents a significant advance, paving the way for new coarse-grained analyses of such complex systems.

1.12

I would like to commend the authors once again for the extensive compilation of test cases and for their commitment to making both the methodology and the benchmark dataset publicly available. This represents a valuable and commendable contribution to the field.

(Remarks on code availability)

Version 3:

Reviewer comments:

Reviewer #1

(Remarks to the Author)

The authors have satisfactorily addressed most of my concerns.

(Remarks on code availability)
